# Mather *β*-Function for Ellipses and Rigidity

**DOI:** 10.3390/e24111600

**Published:** 2022-11-03

**Authors:** Michael Bialy

**Affiliations:** School of Mathematical Sciences, Raymond and Beverly Sackler Faculty of Exact Sciences, Tel-Aviv University, Tel Aviv 6997801, Israel; bialy@tauex.tau.ac.il

**Keywords:** Birkhoff billiard, invariant curve, action minimizers: rigidity, integrable billiards

## Abstract

The goal of the first part of this note is to get an explicit formula for rotation number and Mather β-function for ellipse. This is done here with the help of non-standard generating function of billiard problem. In this way the derivation is especially simple. In the second part we discuss application of Mather β-function to rigidity problem.

## 1. Introduction

Consider the confocal family of ellipses
$$E_{\lambda }=\left\{\frac{x^{2}}{a^{2}-\lambda }+\frac{y^{2}}{b^{2}-\lambda }=1\right\},\;0<\lambda <b^{2}<a^{2}.$$

The initial ellipse is $E=E_{0}$. Polygonal lines with the vertices on *E* circumscribed about confocal caustic $E_{\lambda }$ correspond to billiard trajectories of the billiard in *E*. A caustic $E_{\lambda }$ is called rational [1], of rotation number ρ=m/n, if a billiard trajectory circumscribing $E_{\lambda }$ closes after *n* reflections making *m* rotations. These closed billiard trajectories are called Poncelet polygons. By famous Poncelet theorem if one billiard trajectory tangent to $E_{\lambda }$ is closed with ρ=m/n, then all of them are closed with the same ρ. Given a caustic $E_{\lambda }$, all Poncelet polygons have the same perimeter. Mather β-function assigns to the rotation number ρ of $E_{\lambda }$ the value of this perimeter divided by the number of vertices. Let me remark that traditionally Mather β-function is negative of ours. However we prefer, for convenience, sign + for generating function and hence for Mather β-function as well.

**Example** **1.**
*It is not difficult to compute the perimeter and the corresponding λ for *4*-gons. Namely the perimeter equals $4\sqrt{a^{2}+b^{2}}$ and hence $\beta \left(1/4\right)=\sqrt{a^{2}+b^{2}}$, $\sqrt{\lambda }=ab/\sqrt{a^{2}+b^{2}}$. The perimeter of Poncelet triangles and the corresponding λ can be geometrically found, but this requires solution of cubic equation. We leave this as an exercise.*


In this note we show how to compute the perimeter of the Poncelet polygons for a given caustic $E_{\lambda }$. By a different method a similar formula was discovered in a recent paper [2]. Notice that the straightforward computation of the lengths of the edges seems to be difficult. The main idea of this paper is to use *non-standard generating function* of the billiard. This function was found in [3] for ellipsoids, and in [4,5] for general convex billiard tables. Using this idea we replace the straightforward computation of the action functional by expressing this functional via the non-standard generating function. This approach leads immediately to a formula containing pseudo-elliptic integral, which can be further reduced to elliptic integrals, using [6]. The question of existence of such a formula was explicitly addressed by S. Zelditch in [7,8]. We also get by our method the known formulas for rotation number and the invariant measure [1,9] in a very direct way. In addition, we derive in Section 6 a simple formula for the so-called Lazutkin parameter of the caustic $E_{\lambda }$.

There is an extensive literature on Poncelet porism, formulas for invariant measure and the rotation number. I refer to the incomplete list of papers on the subject [1,9,10,11,12,13].

The non-standard generating function for convex billiards has been already used in our paper [14], explaining conservation laws for elliptical billiards discovered recently by Dan Reznik [15,16] et al., see also [11,12,17]. Additionally, the non-standard generating function is a key ingredient in the recent proof of a part of Birkhoff conjecture for centrally symmetric billiard tables [18].

Mather β-function is very important function related both to classical dynamics inside the domain as well as to the spectral problems. In this paper, we shall discuss in Section 8 the relation of Mather β-function to the rigidity questions. The idea to use Mather β-function for rigidity in billiards belongs to K.F. Siburg [19]. We refer [1,8,20,21,22] for further developments and other approaches.

## 2. Results

In this section, we formulate our main contributions. Other results are placed in the corresponding sections.

**Theorem** **1.**
*Consider the invariant curve of rotation number ρ corresponding to the caustic $E_{\lambda }$. Mather β-function corresponding to the caustic $E_{\lambda }$ is given by the following formula:*

$$\beta \left(\rho \right)=\frac{2ce\sqrt{e^{2}-f^{2}}}{e^{2}-1}-\frac{2cf}{K(k)}\left[K\left(k\right)E\left(\phi ,k\right)-E\left(k\right)F\left(\phi ,k\right)\right],\;k=1/f,\;\phi =\arcsin\frac{\sqrt{\lambda }}{b},$$

*where E(ϕ,k) is elliptic integral of the second kind, K(k),E(k) are complete elliptic integrals of first and second kind, and e,f are eccentricities of the ellipses $E,E_{\lambda }$.*


**Corollary** **1.**
*The following formula holds*

$$\beta \left(\rho \right)=\frac{2a\sqrt{\lambda }}{b}-2\sqrt{a^{2}-\lambda }E\left(\phi ,k\right)+\rho \left|E_{\lambda }\right|,$$


$$\phi =\arcsin\frac{\sqrt{\lambda }}{b},\;k=1/f,$$

*where $|E_{\lambda }|$ denotes the circumference of the ellipse $E_{\lambda }$.*


We give a proof of these formulas Section 1.

**Example** **2.**
*(1) One can see from this formula that for ρ=0, that is when f→e (confocal ellipse coincides with the boundary, i.e., λ=0), it follows that ϕ→0 and hence β→0.*

*(2) When f→1 (corresponding to the confocal ellipse shrinking to the segment between the focii), β→2a—the diameter (only the first summand of the formula remains, the second one tends to zero).*


We shall discuss now the relation of Mather β-function to the rigidity questions. The following problem is important. Let $\Omega_{1},\Omega_{2}$ be two strictly convex domains having the same Mather β-functions $\beta_{1}=\beta_{2}$, can one state that the domains are isometric. It is especially important in view of its applications to spectral rigidity.

Remarkably, if $\Omega_{1}$ is an ellipse then there are many approaches leading to the affirmative answer. In this paper, we do not consider infinitesimal behavior of Mather β-function at 0 (cf. [21,22]), but rather study this function on a finite neighborhood of 0. Our contribution is based on the recent paper with a partial resolution of Birkhoff conjecture for centrally symmetric convex billiards [18]. The result of [18] can be formulated in terms of Mather β-function as follows:

**Theorem** **2.**
*Let $\Omega_{1},\Omega_{2}$ be two strictly convex $C^{2}$-smooth centrally symmetric planar domains such that $\Omega_{1}$ is an ellipse. Suppose that Mather β-functions $\beta_{1},\beta_{2}$ of the domains satisfy*

$$\beta_{1}\left(\rho \right)=\beta_{2}\left(\rho \right),\;\forall \rho \in \left(0,\frac{1}{4}\right].$$

*Then $\Omega_{2}$ is an ellipse isometric to $\Omega_{1}$.*


In Section 8, we shall give the proof of this result and discuss further application of Mather β-function to rigidity problems.

## 3. Preliminaries and Methods

### 3.1. Non-Standard Generating Function

Consider the space of oriented lines in the plane $R^{2}\left(x,y\right)$. A line can be written as
cosφ·x+sinφ·y=p,
where φ is the direction of the right normal to the oriented line, and *p* is the signed distance to the line from the origin. Thus, (p,φ) are coordinates in the space of oriented lines, see Figure 1. The 2-form ρ=dp∧dφ is the area (symplectic) form on the space of oriented lines used in geometrical optics and integral geometry.

Consider a smooth strictly convex billiard curve γ, and let h(φ) be its support function, that is, the distance from the origin (supposed to be inside γ) to the tangent line to γ at the point where the outer normal has direction φ. The sub-space A of the oriented lines intersecting the curve γ is the phase space cylinder of the billiard map. The billiard transformation acts on A as an exact symplectic map.
$$T:\left(p_{1},\varphi_{1}\right)\mapsto \left(p_{2},\varphi_{2}\right)$$
sends the incoming trajectory to the outgoing one. Let
$$\psi =\frac{\varphi_{1}+\varphi_{2}}{2},\;\delta =\frac{\varphi_{2}-\varphi_{1}}{2},$$
where ψ is the direction of the outer normal at the reflection point and δ is the reflection angle.

**Proposition** **1.**
*The function*

$$S\left(\varphi_{1},\varphi_{2}\right)=2h\left(\frac{\varphi_{1}+\varphi_{2}}{2}\right)\sin\left(\frac{\varphi_{2}-\varphi_{1}}{2}\right)=2h\left(\psi \right)\sin\delta$$

*is a generating function of the billiard transformation, that is, $T\left(p_{1},\varphi_{1}\right)=\left(p_{2},\varphi_{2}\right)$ if and only if*

$$-\frac{\partial S_{1}\left(\varphi_{1},\varphi_{2}\right)}{\partial \varphi_{1}}=p_{1},\;\frac{\partial S_{2}\left(\varphi_{1},\varphi_{2}\right)}{\partial \varphi_{2}}=p_{2}.$$



**Proof.** We refer to Figure 2.One has
$$-\frac{\partial S_{1}\left(\varphi_{1},\varphi_{2}\right)}{\partial \varphi_{1}}=-h^{\prime }\left(\psi \right)\sin\delta +h\left(\psi \right)\cos\delta .$$
The position vector of the point of the curve γ with the outer normal having direction ψ is
$$\gamma \left(\psi \right)=h\left(\psi \right)\left(\cos\psi ,\sin\psi \right)+h^{\prime }\left(\psi \right)\left(-\sin\psi ,\cos\psi \right)$$
(this formula is well known in convex geometry). Then, using some trigonometry,
$$p_{1}=\gamma \left(\psi \right)\cdot \left(\cos\varphi_{1},\sin\varphi_{1}\right)=h\left(\psi \right)\cos\delta -h^{\prime }\left(\psi \right)\sin\delta ,$$
as needed. The argument for $p_{2}$ is similar. □

**Figure 2 entropy-24-01600-f002:**
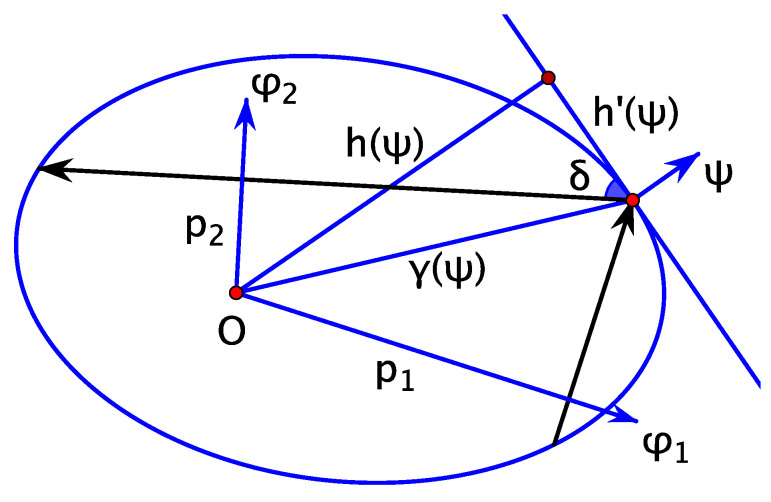
To Proposition 1.

In order to use the function *S* for ellipse let me remind the computation of the support function, with respect to the center of the ellipse, as a function of ψ which is the angle made by the outer normal with the positive *x*-axes.

**Lemma** **1.**
*Let E be the ellipse $\left\{\frac{x^{2}}{a^{2}}+\frac{y^{2}}{b^{2}}=1\right\}$. One has:*

$$h\left(\psi \right)=\sqrt{a^{2}\cos^{2}\psi +b^{2}\sin^{2}\psi }.$$



**Proof.** Consider a point (ξ,η) of the ellipse. A normal vector is given by
$$N=\left(\frac{\xi }{a^{2}},\frac{\eta }{b^{2}}\right)=\ell \left(\cos\psi ,\sin\psi \right),\;\ell =\left|N\right|.$$
and the tangent line at this point has the equation
$$\frac{\xi x}{a^{2}}+\frac{\eta y}{b^{2}}=1.$$
The distance from the origin to this line is
$$\frac{1}{\sqrt{\frac{\xi^{2}}{a^{4}}+\frac{\eta^{2}}{b^{4}}}}=\frac{1}{\ell }.$$On the other hand,
$$\xi =a^{2}\ell \cos\psi ,\;\eta =b^{2}\ell \sin\psi ,$$
and the equation of the ellipse implies that
$$\ell^{2}=\frac{1}{a^{2}\cos^{2}\psi +b^{2}\sin^{2}\psi }.$$
Therefore $h\left(\psi \right)=1/\ell =\sqrt{a^{2}\cos^{2}\psi +b^{2}\sin^{2}\psi }$, as claimed. □

### 3.2. Integral for Elliptic Billiard in Various Forms

Billiard in ellipse is integrable. The integral can be understood at least in three ways.

1. Jacobi-Chasles integral λ.

Given an oriented line not intersecting the segment between the focii. Consider the confocal ellipse
$$E_{\lambda }=\left\{\frac{x^{2}}{a^{2}-\lambda }+\frac{y^{2}}{b^{2}-\lambda }=1\right\}$$
tangent to this line then λ is an integral of the billiard, i.e., it remains constant under the reflections.

2. Joachimsthal integral $J:=\frac{\sin\delta }{h}$.

This corresponds to the conservation of <v,∇q>, where *v* the unit vector of the line, and *q* is the quadratic form $q\left(x\right)=\frac{<Qx,x>}{2}$, with the diagonal matrix $Q=diag(\frac{1}{a^{2}},\frac{1}{b^{2}})$, see Figure 3.

Indeed, the conservation follows from the following implications:$$<Q\left(x+x^{\prime }\right),x^{\prime }-x>=0\;\Rightarrow \;<Qx,v>=-<Qx^{\prime },v>.$$
Additionally,
$$<Qx^{\prime },v+v^{\prime }>=0\;\Rightarrow \;<Qx^{\prime },v^{\prime }>=-<Qx^{\prime },v>\Rightarrow <Qx,v>=<Qx^{\prime },v^{\prime }>.$$
One can express this conservation law in terms of h,δ as follows:$$-<v,\nabla f>=-|\nabla f|\sin\delta =|Qx|\sin\delta =\frac{\sin\delta }{h}=J.$$
Here we used that $\left|Qx\right|=\frac{1}{h}$ as we explained in the proof of Lemma 1.

3. Product of two momenta *F*.

Let us consider a segment of the billiard trajectory tangent to a confocal ellipse $E_{\lambda }$ with the semi-axes $\sqrt{b^{2}-\lambda }<\sqrt{a^{2}-\lambda }$. Let $d_{1},d_{2}$ be the distances from the foci to the line and α be the direction of its normal. Then we define $F:=d_{1}d_{2}$ (This definition of the integral we learned from Michael Berry). It then follows from the next theorem that *F* is indeed an integral.

### 3.3. The Relations between Conserved Quantities

**Theorem** **3.**
*The following relations hold true:*
*1.* 

$J=\sqrt{\lambda }/ab$

*2.* 

$F=\sqrt{b^{2}-\lambda }$

*3.* 
*In terms of the eccentricities $e=\frac{a}{c},f=\frac{\sqrt{a^{2}-\lambda }}{c}$ of $E,E_{\lambda }$ we have the formulas:*

$$\lambda =c^{2}\left(e^{2}-f^{2}\right),\;J=\frac{\sqrt{e^{2}-f^{2}}}{c\;e\sqrt{e^{2}-1}},\;where\;c=\sqrt{a^{2}-b^{2}}.$$




**Proof.** (1) Consider an oriented line passing through the point (a,0) with right normal having angle δ (see Figure 4). Then for *p* of this line we have
$$p=\sqrt{\left(a^{2}-\lambda \right)\cos^{2}\delta +\left(b^{2}-\lambda \sin^{2}\delta \right)},$$
and hence
p=acosδ,
see Figure 4. Therefore these two give
$$b\;\sin\delta =\sqrt{\lambda }.$$On the other hand from the definition of *J* we have:
$$J=\frac{\sin\delta }{a}.$$
Thus,
$$J=\frac{\sqrt{\lambda }}{ab}.$$(2) Given a line with coordinates (p,φ), we have
$$d_{1}=p-c\cos\phi ,\;d_{2}=p+c\cos\varphi ,$$
where $c^{2}=a^{2}-b^{2}$ (see Figure 5).(1)$$F:=d_{1}d_{2}=p^{2}-c^{2}\cos^{2}\varphi .$$
If the line is tangent to $E_{\lambda }$, then $p=h_{\lambda }\left(\varphi \right)$, where $h_{\lambda }$ is the support function of $E_{\lambda }$. Hence, using Lemma 1 we rewrite (Equation 1)
$$F=\left(a^{2}-\lambda \right)\cos^{2}\varphi +\left(b^{2}-\lambda \right)\sin^{2}\varphi -c^{2}\cos^{2}\varphi =b^{2}-\lambda .$$
(3) Follows from item (1) and the definition of eccentricities. □

## 4. Invariant Measure on an Invariant Curve

Suppose we have a curve on the phase cylinder A which is invariant under the billiard map *T*. Suppose this curve is a graph and lies in the level set of the integral F(p,φ)=const. Then there is a natural measure dμ on the curve which is invariant under *T*. According to V.I. Arnold this is called Gelfand-Leray form, which by another Arnold’ principle was probably discovered earlier. Next we compute this measure explicitly.

**Theorem** **4.**
*The invariant measure on the invariant curve corresponding to the value J of Joachimsthal integral and other related quantities given by Theorem 3 is given by the formula:*

$$d\mu =\frac{d\psi }{\sqrt{a^{2}-c^{2}\sin^{2}\psi }\sqrt{\left(1-J^{2}a^{2}\right)+J^{2}c^{2}\sin^{2}\psi }}.$$

*Therefore the measure of the arc [0,ψ] equals*

$$\mu \left(\left[0,\psi \right]\right)=\frac{1}{cf}F\left(\varphi ,\frac{1}{f}\right),\;\varphi =\arcsin\sqrt{\frac{\left(d+1\right)\tan^{2}\psi }{\left(d+1\right)\tan^{2}\psi +d}},\;$$


$$d=\frac{1-J^{2}a^{2}}{J^{2}c^{2}}=\frac{\left(b^{2}-\lambda \right)e^{2}}{\lambda }=\frac{f^{2}-1}{e^{2}-f^{2}}e^{2}>0$$

*The measure of the whole invariant curve equals*

$$U=\frac{4}{cf}F\left(\frac{\pi }{2},\frac{1}{f}\right).$$

*Here and below e,f are the eccentricities of $E,E_{\lambda }$ and $F\left(\varphi ,k\right)=\int_{0}^{\varphi }\frac{dt}{\sqrt{1-k^{2}\sin^{2}t}}$ is the elliptic integral of the first kind.*


**Proof.** The invariant measure on the curve {F=const} can be written as:
$$d\mu =\frac{1}{F_{p}}d\varphi .$$
Due to explicit form of *F* in (p,φ) coordinates (Theorem 3) we have:
$$d\mu =\frac{1}{p}d\varphi$$
We compute using the change of variable on the invariant curve φ→ψ (see Figure 6):
φ=ψ+δ(ψ).Using the formula
$$p=h\left(\psi \right)\cos\delta \left(\psi \right)+h^{\prime }\left(\psi \right)\sin\delta \left(\psi \right)$$
We have
(2)$$d\mu =\frac{1}{p}d\varphi =\frac{(1+\delta^{\prime }\left(\psi \right))d\psi }{h\left(\psi \right)\cos\delta \left(\psi \right)+h^{\prime }\left(\psi \right)\sin\delta \left(\psi \right)}$$Next we use the explicit form of Joachimsthal integral:
Jh(ψ)=sinδ(ψ)
and hence also
$$Jh^{\prime }\left(\psi \right)=\cos\delta \left(\psi \right)\delta^{\prime }\left(\psi \right).$$
Substituting into (Equation 2) we get
(3)$$d\mu =\frac{1}{p}d\varphi =\frac{J(1+\delta^{\prime }\left(\psi \right))d\psi }{\sin\delta \left(\psi \right)\cos\delta \left(\psi \right)(1+\delta^{\prime }\left(\psi \right))}=\frac{Jd\psi }{\sin\delta (\psi )\cos\delta (\psi )}=\frac{d\psi }{h\sqrt{1-h^{2}}}.$$
thus we compute using Lemma 1:
$$d\mu =\frac{d\psi }{\sqrt{a^{2}\cos^{2}\psi +b^{2}\sin^{2}\psi }\sqrt{1-J^{2}\left(a^{2}\cos^{2}\psi +b^{2}\sin^{2}\psi \right)}}=$$
$$\frac{d\psi }{\sqrt{a^{2}-\left(a^{2}-b^{2}\right)\sin^{2}\psi }\sqrt{\left(1-J^{2}a^{2}\right)+J^{2}\left(a^{2}-b^{2}\right)\sin^{2}\psi }}=$$
$$\frac{d\psi }{\sqrt{a^{2}-c^{2}\sin^{2}\psi }\sqrt{\left(1-J^{2}a^{2}\right)+J^{2}c^{2}\sin^{2}\psi }}.$$
Therefore the measure of the segment [0,ψ]
$$\mu \left(\left[0,\psi \right]\right)=\int_{0}^{\psi }\frac{d\psi }{\sqrt{a^{2}-c^{2}\sin^{2}\psi }\sqrt{\left(1-J^{2}a^{2}\right)+J^{2}c^{2}\sin^{2}\psi }}.$$
Changing to $x=\sin^{2}\psi$ we get
$$\mu \left(\left[0,\psi \right]\right)=\frac{1}{2}\int_{0}^{\sin^{2}\psi }\frac{dx}{\sqrt{x(1-x)}\sqrt{a^{2}-c^{2}x}\sqrt{\left(1-J^{2}a^{2}\right)+J^{2}c^{2}x}}=$$
$$\frac{1}{2Jc^{2}}\int_{0}^{\sin^{2}\psi }\frac{dx}{\sqrt{x(1-x)}\sqrt{e^{2}-x}\sqrt{d+x}}=\frac{g}{2Jc^{2}}F\left(\varphi ,k\right),$$
where in the last step we used the reduction of the pseudo-elliptic integral to the elliptic integral of the first kind [6] (p. 112; integral 254.00). In the last formula e=a/c is the eccentricity of the ellipse, $c=\sqrt{a^{2}-b^{2}},d=\frac{1-J^{2}a^{2}}{J^{2}c^{2}}$. Now we need to compute parameters of the elliptic integral given in [6] (p. 112). In order to compute *d* we use Theorem 3
(4)$$d=\frac{1-J^{2}a^{2}}{J^{2}c^{2}}=\frac{\left(b^{2}-\lambda \right)e^{2}}{\lambda }=\frac{f^{2}-1}{e^{2}-f^{2}}e^{2}>0,$$
$$\;J=\frac{\sqrt{e^{2}-f^{2}}}{ce\sqrt{e^{2}-1}},\;\lambda =c^{2}\left(e^{2}-f^{2}\right),$$
where *f* is the eccentricity of $E_{\lambda }$. Moreover, we compute the parameters k,φ,g for the elliptic integral.
$$k=\sqrt{\frac{e^{2}+d}{e^{2}\left(1+d\right)}}=1/f,\;g=\frac{2}{\sqrt{e^{2}\left(1+d\right)}}=\frac{2\sqrt{e^{2}-f^{2}}}{fe\sqrt{e^{2}-1}}.$$
The angle φ is computed by the formula:
(5)$$\;\sin^{2}\varphi =\frac{\left(d+1\right)\left(\sin^{2}\psi \right)}{\sin^{2}\psi +d}=\frac{\left(d+1\right)\tan^{2}\psi }{\left(d+1\right)\tan^{2}\psi +d},$$
where *d* is given in (Equation 4). Next we see that the coefficient
$$\frac{g}{2Jc^{2}}=\frac{1}{cf}.$$
Thus, finally we have
$$\mu \left(\left[0,\psi \right]\right)=\frac{1}{cf}F\left(\varphi ,\frac{1}{f}\right),\;U=\frac{4}{cf}F\left(\frac{\pi }{2},\frac{1}{f}\right),$$
where *U* is the measure of the whole curve. □

## 5. Mather *β*-Function

Now we are in position to find Mather β-function for ellipse stated in Theorem 1 and Corollary 1. We shall use the invariant measure and non-standard generating function *S*. Consider the invariant curve of the rotation number $\rho =\frac{m}{n}$ corresponding to the rational caustic $E_{\lambda }$ and to the value *J* of Joachimsthal integral. We shall give a proof of the formula for rational rotation number ρ, but it is easy to see that it remains valid for irrational ρ.

**Proof of Theorem 1.** Let ρ=m/n and $(p_{i},\varphi_{i}),i=1,\ldots ,n$ denote the coordinates of the edges $l_{i}$ of a Poncelet polygon. Set
$$\psi_{i}=\frac{\varphi_{i-1}+\varphi_{i}}{2},\;\delta_{i}=\frac{\varphi_{i}-\varphi_{i-1}}{2}.$$The perimeter of the Poncelet polygon can be computed by means of the generating function *S* given in Proposition 1 as follows (see [14]):
(6)$$\beta \left(\frac{m}{n}\right)=\frac{1}{n}\sum_{i=1}^{n}S\left(\varphi_{i-1},\varphi_{i}\right)=\frac{2}{n}\sum_{i=1}^{n}h\left(\psi_{i}\right)\sin\delta_{i},$$Next we integrate both sides of (Equation 6) with respect to the measure dμ and using the invariance of the measure we get:
$$\beta \left(\frac{m}{n}\right)U=2\int h\left(\psi \right)\sin\delta \left(\psi \right)d\mu ,$$
where *U* is the measure of the whole curve. Thus, we have using the explicit expression of the measure (3):
$$\beta \left(\rho \right)=\frac{2}{U}\int_{0}^{2\pi }\frac{Jh\sin\delta }{\sin\delta \cos\delta }d\psi =\frac{2J}{U}\int_{0}^{2\pi }\frac{h}{\sqrt{1-J^{2}h^{2}}}d\psi .$$
Substitute the explicit formula for *h* we obtain:
$$\beta \left(\rho \right)=\frac{8J}{U}\int_{0}^{\pi /2}\frac{\sqrt{a^{2}\cos^{2}\psi +b^{2}\sin^{2}\psi }}{\sqrt{1-J^{2}\left(a^{2}\cos^{2}\psi +b^{2}\sin^{2}\psi \right)}}d\psi =$$
$$\frac{8J}{U}\int_{0}^{\pi /2}\frac{\sqrt{a^{2}-c^{2}\sin^{2}\psi }}{\sqrt{1-J^{2}a^{2}+J^{2}c^{2}\sin^{2}\psi }}d\psi =$$
$$\frac{8Jc}{UJc}\int_{0}^{\pi /2}\frac{\sqrt{e^{2}-\sin^{2}\psi }}{\sqrt{d+\sin^{2}\psi }}d\psi =\frac{8}{U}\int_{0}^{\pi /2}\frac{\sqrt{e^{2}-\sin^{2}\psi }}{\sqrt{d+\sin^{2}\psi }}\frac{d\sin^{2}\psi }{2\sin\psi \cos\psi }=$$
$$\frac{4}{U}\int_{0}^{1}\frac{\sqrt{e^{2}-x}}{\sqrt{d+x}\sqrt{x(1-x)}}dx=\frac{4e^{2}g}{U}\left(\frac{k^{2}}{\alpha^{2}}F\left(\pi /2,k\right)-\left(\frac{k^{2}}{\alpha^{2}}-1\right)\Pi \left(\pi /2,\alpha^{2},k\right)\right).$$
where we used the values $\alpha^{2}=\frac{1}{1+d},\;k=\frac{1}{f}$ and g,φ as above. This reduction to the complete elliptic integral of the third kind is given in [6] (p.112 integral 254.13 then 339.01). Next we use [6] (integral 414.01) for the complete integral $\Pi \left(\pi /2,\alpha^{2},k\right)=:\Pi \left(\alpha^{2},k\right)$ and finally obtain:
$$\beta \left(\rho \right)=\frac{4e^{2}g}{U}\left(\frac{k^{2}}{\alpha^{2}}K\left(k\right)-\left(\frac{k^{2}-\alpha^{2}}{\alpha^{2}}\right)\left(K\left(k\right)+\frac{\alpha [K(k)E(\phi ,k)-E(k)F(\phi ,k)]}{\sqrt{\left(1-\alpha^{2}\right)\left(k^{2}-\alpha^{2}\right)}}\right)\right),$$
where ϕ=arcsin(α/k). Simplifying we get:
$$\beta \left(\rho \right)=\frac{4e^{2}g}{U}\left(K\left(k\right)-\left(\frac{\sqrt{k^{2}-\alpha^{2}}}{\alpha \sqrt{1-\alpha^{2}}}\right)\left[K\left(k\right)E\left(\phi ,k\right)-E\left(k\right)F\left(\phi ,k\right)\right]\right).$$
Substituting the values of parameters
$$g=\frac{2\sqrt{e^{2}-f^{2}}}{fe\sqrt{e^{2}-1}},\;U=\frac{4}{cf}K\left(k\right),\;k=1/f,\;\alpha^{2}=\frac{e^{2}-f^{2}}{f^{2}\left(e^{2}-1\right)},\;k^{2}-\alpha^{2}=\frac{f^{2}-1}{f^{2}\left(e^{2}-1\right)}$$
we get:
$$\beta \left(\rho \right)=\frac{2ce\sqrt{e^{2}-f^{2}}}{e^{2}-1}-\frac{2cf}{K(k)}\left[K\left(k\right)E\left(\phi ,k\right)-E\left(k\right)F\left(\phi ,k\right)\right],$$
$$\phi =\arcsin\sqrt{\frac{e^{2}-f^{2}}{e^{2}-1}}=\arcsin\frac{\sqrt{\lambda }}{b}.$$□

**Proof of the Corollary 1.** This follows immediately from Theorem 1 using the following relations. The first is on the perimeter of the ellipse $|E_{\lambda }|$, and the second for the rotation number ρ which we shall prove in Theorem 2 in Section 7.
$$|E_{\lambda }|=4\int_{0}^{\pi /2}\sqrt{\left(a^{2}-\lambda \right)-c^{2}\sin^{2}t}\;dt=4\sqrt{a^{2}-\lambda }\;E\left(k\right),\;k=1/f,\;cf=\sqrt{a^{2}-\lambda },$$
where *f* is the eccentricity of $E_{\lambda }$ as above.
$$\;\rho =\frac{F(\phi ,k)}{2K(k)}.$$□

## 6. Mather *β*-Function and the Lazutkin Parameter

Let me remind the notion of the Lazutkin parameter. Given a convex caustic C of convex billiard curve γ (not necessarily ellipse), one has a conservation law stating that for any point P∈γ the Lazutkin parameter
$$L:=\left|PX|+|PY|-\right|\overset{⌢}{XY}|$$
does not depend on the point *P* (see [23]). Here X,Y∈C are the tangency points of tangent lines to C from *P* and overarc denotes the arc between the indicated points.

Suppose $P_{i},i=1,\ldots ,n$ are the vertices of billiard *n*-periodic trajectory P making *m* turns. For any vertex $P_{i}$ we write the Lazutkin parameter:$$L=|P_{i}X_{i}\left|+\right|P_{i}Y_{i}\left|-\right|\overset{⌢}{X_{i}Y_{i}}|,\;i=1,\ldots ,n.$$
Summing these identities we get
nL=|P|−m|C|.
Dividing by *n*, we obtain the general formula (see [19]), valid for any billiard with convex caustic C:β(ρ)=L+ρ|C|,
where *L* is Lazutkin parameter, $\rho =\frac{m}{n}$ is the rotation number and |·| is the perimeter. Comparing the last formula with one of Corollary 1 we get the following:

**Corollary** **2.**
*For the Lazutkin parameter L of the caustic $E_{\lambda }$ of the elliptic billiard E we have the following formula:*

$$L\left(E_{\lambda }\right)=\frac{2a\sqrt{\lambda }}{b}-2\sqrt{a^{2}-\lambda }\;E\left(\phi ,k\right).$$



## 7. Rotation Number *ρ*

In this section, we give another derivation of the formula for the rotation number ρ corresponding to caustic $E_{\lambda }$ [1,9].

**Theorem** **5.**
*For the invariant curve corresponding to caustic $E_{\lambda }$ having eccentricity f the rotation number is:*

$$\rho =\frac{F(\phi ,k)}{2K(k)},\;k=\frac{1}{f},\;\phi =\arcsin\sqrt{\frac{e^{2}-f^{2}}{e^{2}-1}}=\arcsin\frac{\sqrt{\lambda }}{b},$$

*where K(k)=F(π/2,k) is the complete elliptic integral.*


**Example** **3.**
*We see from this Theorem that for λ→0 the ϕ→0 and hence ρ→0. On the other hand if λ→b that is f→1 we have ϕ→π/2 and hence ρ→1/2.*


**Remark** **1.**
*This formula is given in [1,9]. A beautiful method to get formula for rotation number is given in [24]. Unfortunately there is a computational mistake for the integrals at the end of page 298. Another formula for the rotation number is given without proof in [13]. However, in that formula f→1 does not imply to ρ→1/2.*


**Proof of Theorem 5.** We shall use the formula for rotation number:
ρ=μ[ψ,T(ψ)]/U,
where ψ is a point on the curve and T(ψ) its image. This is independent on the choice of ψ since measure μ is invariant. Here and below we use ψ as a coordinate on the invariant curve related to the angle φ by the formula φ=ψ+δ(ψ) as before. Now we shall choose ψ in this formula in such a way that the segment [ψ,T(ψ)] is vertical and tangent to $E_{\lambda }$ (see Figure 7):
ψ=−θandT(ψ)=θ.We can easily compute θ using the normal vector $N=\left(\frac{x}{a^{2}},\frac{y}{b^{2}}\right),$ where by the definition of θ, we have
$$x=\sqrt{a^{2}-\lambda },\;y=\frac{\sqrt{\lambda }b}{a}.$$
Hence we get:
(7)$$\tan\theta =\frac{a^{2}}{b^{2}}\frac{\sqrt{\lambda }b}{a\sqrt{a^{2}-\lambda }}=\frac{a}{b}\frac{\sqrt{\lambda }}{\sqrt{a^{2}-\lambda }}=\frac{e\sqrt{e^{2}-f^{2}}}{f\sqrt{e^{2}-1}}.$$
It then follows from Theorem 4 that
$$\mu \left[-\theta ,\theta \right]=2\mu \left[0,\theta \right]=\frac{2}{cf}F\left(\phi ,\frac{1}{f}\right),$$
where
$$\;\sin^{2}\phi =\frac{\left(d+1\right)\tan^{2}\theta }{\left(d+1\right)\tan^{2}\theta +d}.$$
Substituting d,d+1 from (Equation 1) and tanθ from (Equation 7) we obtain:
$$\sin\phi =\sqrt{\frac{e^{2}-f^{2}}{e^{2}-1}}=\frac{\sqrt{\lambda }}{b}.$$
Thus, we have for the rotation number:
$$\rho =\frac{2}{U}\mu \left[0,\theta \right]=\frac{2}{cfU}F\left(\phi ,\frac{1}{f}\right)=\frac{1}{2F\left(\frac{\pi }{2},\frac{1}{f}\right)}F\left(\phi ,\frac{1}{f}\right),\;$$
$$\phi =\arcsin\sqrt{\frac{e^{2}-f^{2}}{e^{2}-1}}=\arcsin\frac{\sqrt{\lambda }}{b}.$$□

**Remark** **2.**
*Analogously to the proof of Theorem 1 the following relation can be derived:*

(8)
$$\rho =\frac{4}{\pi U}\int_{0}^{\pi /2}\frac{\arcsin\left(J\sqrt{a^{2}\cos^{2}\psi +b^{2}\sin^{2}\psi }\right)\;d\psi }{\sqrt{a^{2}\cos^{2}\psi +b^{2}\sin^{2}\psi }\sqrt{1-J^{2}\left(a^{2}\cos^{2}\psi +b^{2}\sin^{2}\psi \right)}}.$$

*Indeed, by the the following formula holds for (m,n)-periodic:*

$$2\pi m=\sum_{i=1}^{n}2\delta_{i},$$

*because $2\delta_{i}$ is the angle between the edges $l_{i-1}$ and the $l_{i}$. Integrating this with respect to the invariant measure dμ we get:*

2πmU=2n∫δdμ.

*Thus, we have*

$$\rho =\frac{1}{\pi U}\int \delta d\mu =\frac{1}{\pi U}\int_{0}^{2\pi }\frac{J\delta }{\sin\delta \cos\delta }d\psi .$$

*The last integral gives formula (Equation 8). Notice, that unlike Theorem (5), integral (Equation 8), cannot be reduced to elliptic integrals.*


## 8. Mather *β*-Function and Rigidity

We start this section with the proof of Theorem 2.

**Proof.** The first step is based on a combination of several powerful results. By a Theorem of John Mather [25] the function β is differentiable at a rational point ρ if and only if there is an invariant curve consisting of periodic orbits with rotation number ρ. Moreover, all the orbits lying on these invariant curves are action minimizing. It then follows from Aubry-Mather theory and theorem of Mather on differentiability of β-function that there exist invariant curves of all rotation numbers $\rho \in (0,\frac{1}{4}]$, and these curves foliate the domain between the curve for ρ=1/4 and the boundary of the phase cylinder A (see [19] for the argument in the case of circular billiard).Therefore, billiard in $\Omega_{2}$ meets the assumptions of [18] and hence must be an ellipse.The last step is to show that this ellipse is an isometric copy of $\Omega_{1}$. Indeed let $a_{i}>b_{i},i=1,2$ are semi-axes of the two ellipses. First, take the value of the rotation number $\frac{1}{4}$ and use the equality of the β-functions at the value 1/4. This yields
(9)$$a_{1}^{2}+b_{1}^{2}=a_{2}^{2}+b_{2}^{2}.$$
Second, mention that by the definition β(0)=0 holds true for any domain. However, the derivative $\beta^{\prime }\left(0\right)$ gives the circumference of the domain. Therefore, by the assumption of Theorem 2, we have $\beta_{1}^{\prime }\left(0\right)=\beta_{2}^{\prime }\left(0\right)$ and hence $|\Omega_{1}\left|=\right|\Omega_{2}|$, where |Ω| is the circumference of Ω. Next we use classical formula for |Ω| of arbitrary convex domain via the support function:
$$\left|\Omega \right|=\int_{0}^{2\pi }h\left(\psi \right)d\psi .$$
Therefore for the ellipses $\Omega_{1,2}$ we write
$$|\Omega_{i}|=4\int_{0}^{\pi /2}\sqrt{\frac{a_{i}^{2}+b_{i}^{2}}{2}+\frac{a_{i}^{2}-b_{i}^{2}}{2}\cos2\psi }\;d\psi =$$
$$2\sqrt{2}\int_{0}^{\pi }\sqrt{\left(a_{i}^{2}+b_{i}^{2}\right)+\left(a_{i}^{2}-b_{i}^{2}\right)\cos t}\;\;dt=2\sqrt{2}\int_{0}^{\pi }\sqrt{A+c_{i}^{2}\cos t}\;\;dt,$$
where we introduced $A:=a_{1}^{2}+b_{1}^{2}=a_{2}^{2}+b_{2}^{2}$. Consider now the last integral as a function of the parameter $C:=c^{2}=a^{2}-b^{2}$, while *A* is fixed.
$$f\left(C\right):=2\sqrt{2}\int_{0}^{\pi }\sqrt{A+C\cos t}\;dt$$
Differentiating *f* with respect to *C* we obtain:
$$f^{\prime }=\sqrt{2}\int_{0}^{\pi }\frac{\cos t}{\sqrt{A+C\cos t}}\;dt=\sqrt{2}\int_{0}^{\pi /2}\left[\frac{\cos t}{\sqrt{A+C\cos t}}-\frac{\cos t}{\sqrt{A-C\cos t}}\right]\;dt.$$
It is easy to see that the for t∈(0,π/2) the integrand is negative, hence *f* is strictly monotone decreasing in *C*. Therefore, the equality $|\Omega_{1}\left|=\right|\Omega_{2}|$ is possible only when $C_{1}=C_{2}$. This together with (Equation 9) implies that the ellipses are isometric. □

The second part of the given proof leads naturally to the following.

**Question.** How many values of β-function determine the ellipse in the class of ellipses. More precisely we ask if ellipse is determined by any two values of β-function $\beta \left(\rho_{1}\right),\beta \left(\rho_{2}\right)$ for the rotation numbers $\rho_{1,2}\in \left(0,\frac{1}{2}\right]$.

In order to prove this one needs more analysis of the formula of minimal action of Theorem 1. Notice that in [22] the reconstruction of ellipse is given by means of infinitesimal data of the β-function near 0.

A partial result in the direction of this question is the following

**Theorem** **6.**
*Ellipse can be determined by two values of $\beta \left(\rho_{1}\right),\beta \left(\rho_{2}\right)$ where $\rho_{1}=\frac{1}{2}$ and $\rho_{2}=\frac{m}{n}$ is any rational in $\left(0,\frac{1}{2}\right)$.*


**Proof.** Notice first that $\beta (\frac{1}{2})=2a$ is the diameter of ellipse. We argue by contradiction. Suppose $\Omega_{1},\Omega_{2}$ are two ellipses with the same diameter 2a, satisfying $\beta_{1}\left(\frac{m}{n}\right)=\beta_{2}\left(\frac{m}{n}\right)$, but $b_{1}<b_{2}$, see Figure 8. In this case we can introduce a linear map *A* which is the expansion map along the *y*-axes transforming $\Omega_{1}$ to $\Omega_{2}$. Notice that *A* increases perimeter of any polygon.Denote by $P_{1},P_{2}$ two Poncelet polygons of the rotation number $\frac{m}{n}$ for $\Omega_{1}$ and $\Omega_{2}$, respectively. Obviously, the polygons $A(P_{1})$ and $P_{2}$ have the same rotation number. The condition $\beta_{1}\left(\frac{m}{n}\right)=\beta_{2}\left(\frac{m}{n}\right)$ implies that the perimeters of $P_{1,2}$ are equal:
$$|P_{1}\left|=\right|P_{2}|.$$
Hence we have the inequality
$$|A\left(P_{1}\right)\left|>\right|P_{2}|,$$
since *A* is expanding. However, this contradicts the fact $P_{2}$ is a Poncelet polygon is a length maximizer in its homotopy class. □

**Remark.** It is plausible that the result of Theorem 6 remains valid when the rotation number $\rho_{2}$ is irrational.

## 9. Discussion

Let me pose here most natural problems related the results of this paper:Is it possible to relax symmetry assumption in the main Theorem 1? Our method of proof of Theorem 1 relies on the approach related to the so-called E.Hopf type rigidity phenomenon from [18]. This method is very robust and it is not clear at the moment how it can be generalized.Another problem is to adopt our approach to a smaller neighborhood of the boundary of the phase cylinder.All known approaches to rigidity in billiards, are based on the properties of orbits near the boundary. We believe there are rigidity results based on the behavior far from the boundary.It would be interesting to prove that ellipse is determined by any two values of Mather β function $\beta \left(\rho_{1}\right),\beta \left(\rho_{2}\right)$ where $\rho_{1},\rho_{2}$ are any two rotation numbers in $\left(0,\frac{1}{2}\right)$.

## Figures and Tables

**Figure 1 entropy-24-01600-f001:**
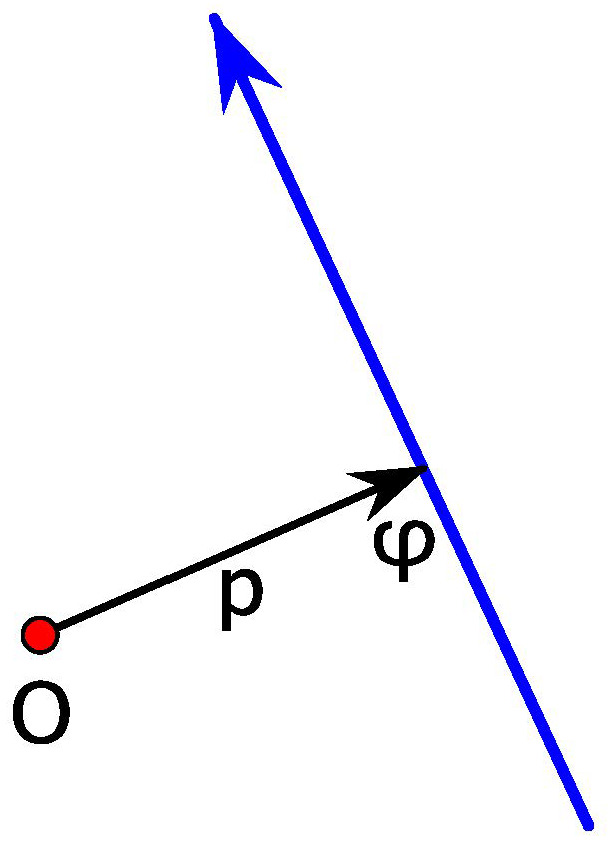
Coordinates in the space of oriented lines.

**Figure 3 entropy-24-01600-f003:**
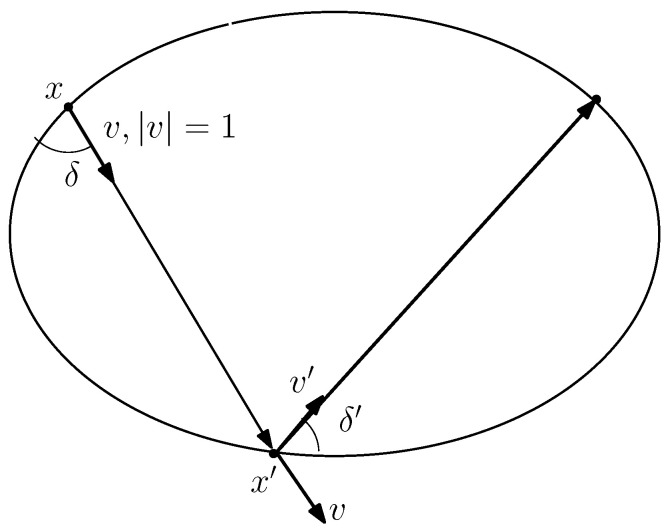
Joachimsthal integral.

**Figure 4 entropy-24-01600-f004:**
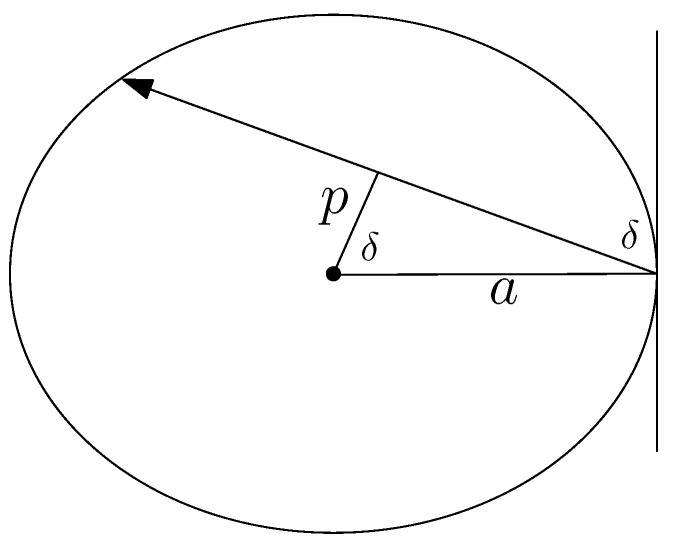
Relation of *J* and λ.

**Figure 5 entropy-24-01600-f005:**
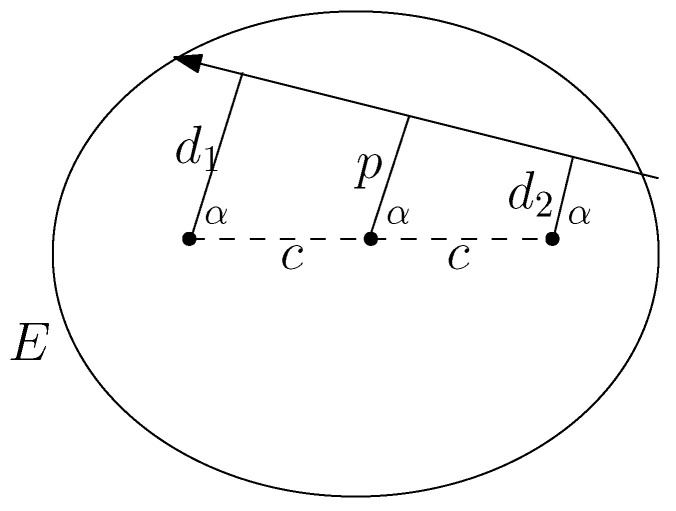
Integral $F=d_{1}d_{2}$.

**Figure 6 entropy-24-01600-f006:**
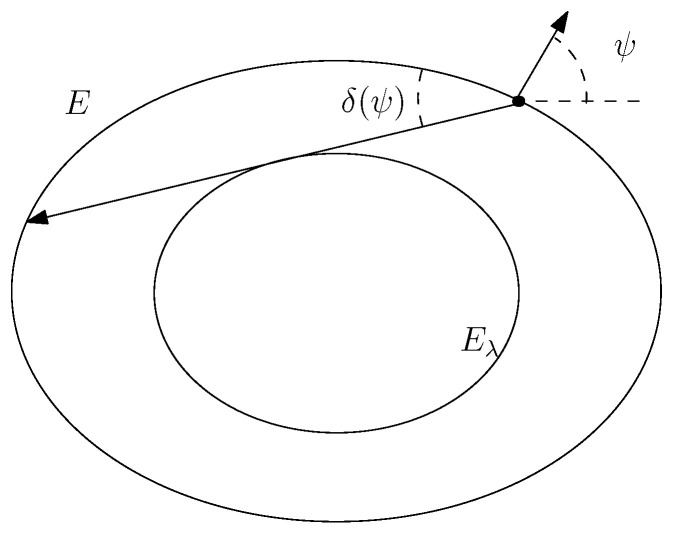
Change of variable on the invariant curve φ→ψ
φ=ψ+δ(ψ).

**Figure 7 entropy-24-01600-f007:**
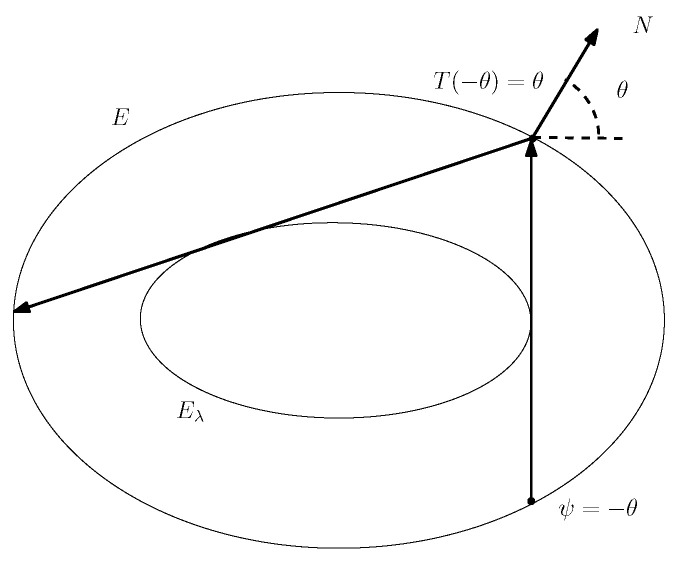
For computation of ρ.

**Figure 8 entropy-24-01600-f008:**
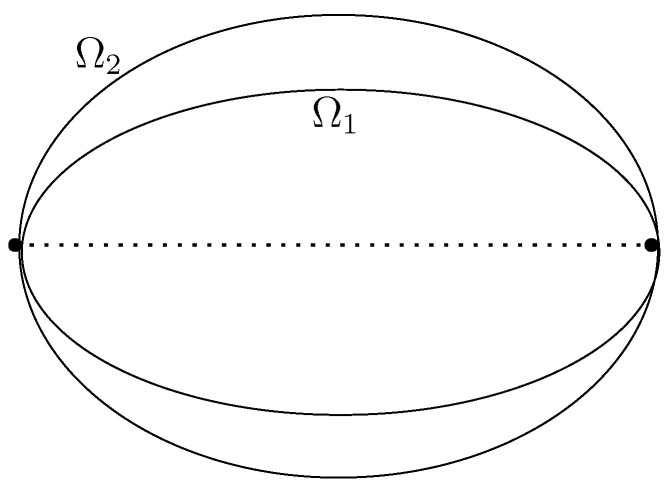
Ellipses with the same diameter.

## Data Availability

Not applicable.

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
