# Peer review of "Mather β-Function for Ellipses and Rigidity"

_entropy, 2022, doi:10.3390/e24111600_

Round 1

Reviewer 1 Report

This paper contains some nice results, and therefore it is publishable However, it makes the impression to be published in a rush. For instance, some notions are used  before they were defined. The style is also rather strange. For instance, the  author does not explain what is rigidity but provides a lot of direct rather elementary computations, including even exact value of derivative of an elementary function. The author is fond to call some results and formulas 'important" and even"remarkable". Such claims need justifications, especially for a special issue of "Quantum Chaos". "Discussion"contains formulations of some intrinsically mathematical problems, rather than explanation why and how the results of further problems are of interest to the readers. Perhaps the Author should elaborate on relation(s) to spectral rigidity.

My impression is that this paper (after being properly rewritten) rather belongs to a mathematical journal. However, it is not for me to decide. It should be between the Author and the Editors.

Some minor remarks.

It seems that the Mather Beta-function was discovered (rather than "independently found" in [14]. 

The paper [13] could not be published at 1844 because the later Prof. Mather was born later.

Reviewer 2 Report

see the attached pdf
